# A Run-Time Reconfiguration Method for an FPGA-Based Electrical Capacitance Tomography System

Damian Wanta * , Waldemar T. Smolik , Jacek Kryszyn , Przemysław Wróblewski and Mateusz Midura

Faculty of Electronics and Information Technology, Warsaw University of Technology, 00-665 Warsaw, Poland; W.Smolik@ire.pw.edu.pl (W.T.S.); J.Kryszyn@ire.pw.edu.pl (J.K.); P.Wroblewski@ire.pw.edu.pl (P.W.); M.Midura@ire.pw.edu.pl (M.M.)
* Correspondence: D.Wanta@ire.pw.edu.pl; Tel.: +48-22-2347577

**Abstract:** A desirable feature of an electrical capacitance tomography system is the adaptation possibility to any sensor configuration and measurement mode. A run-time reconfiguration of a system for electrical capacitance tomography is presented. An original mechanism is elaborated to reconfigure, on the fly, a modular EVT4 system with multiple FPGAs installed. The outlined system architecture is based on FPGA programmable logic devices (Xilinx Spartan) and PicoBlaze soft-core processors. Soft-core processors are used for communication, measurement control and data preprocessing. A novel method of FPGA partial reconfiguration is described, in which a PicoBlaze soft-core processor is used as a reconfiguration controller. Behavioral reconfiguration of the system is obtained by providing run-time access to the program code of a soft-core control processor. The tests using EVT4 hardware and different algorithms for tomographic scanning were performed. A test object was measured using 2D and 3D sensors. The time and resources required for the examined reconfiguration procedure are evaluated.

**Keywords:** data acquisition system; soft-core processor; dynamic reconfiguration; behavioral reconfiguration; impedance tomography





## 1. Introduction

Electrical capacitance tomography (ECT) is an imaging technique enabling the visualization of the spatial distribution of electric permittivity [1]. Due to the significant difference in electric permittivity of liquids and gases, ECT can be successfully applied to monitor dynamic processes in chemical and process engineering [2], such as a slurry flow, multi-phase flow [3,4], batch mixing, separation, combustion [5], etc. The potential for using this technique in medical imaging, e.g., for brain imaging and monitoring lung ventilation, is also being explored [6,7].

In ECT, a cross-sectional image is reconstructed using measurements of the mutual capacitance of electrodes surrounding the examined volume. The number of electrodes, i.e., spatial sampling, is limited due to the extremely low value of measured capacitances, which does not allow for decreasing the electrode area. The ECT weaknesses are low capacitance measurement sensitivity to a permittivity change and poor spatial resolution. High imaging speed, up to several thousand frames per second, the relatively simple design of the sensor and its scalability are ECT strengths.

The ECT sensor itself has a relatively simple structure and can be set up on various installations in laboratory or industrial conditions. The sensor is geometrically scalable, i.e., the measured capacitance does not change with the size of the sensor. Capacitance sensors with different geometries, 2D or 3D configurations and a different number of electrodes and their arrangement can be used [8,9]. The basic measurement procedure is to apply an excitation voltage to one selected electrode and to measure the current using another electrode, which is grounded. Customized electrode excitation strategies for capacitance

measurements can be used for a given sensor configuration, especially using a multichannel device. To improve spatial image resolution of ECT, especially 3D ECT, the number of spatial samples can be increased by physical rotation and shifting of the sensor, or by electronic scanning of electrodes divided into small segments [10]. The reconfiguration of the hardware is required to switch between different electronic scanning strategies [11]. The measurement range is very wide due to the large difference in capacitance value for adjacent and opposing electrode pairs [12]. Run-time gain adjustment in the channels is necessary to improve the signal-to-noise ratio [13]. ECT can be used for imaging objects whose electrical conductivity is not negligible [6]. In this case, electrical permittivity is a complex-valued quantity. The measurement system can register the amplitude and the phase shift of the signal. The real and imaginary part of the permittivity distribution or their sum can be reconstructed.

The ability to customize the measurement sequence to suit any sensor configuration and measurement mode is a desirable feature of a universal ECT system. Field-programmable gate array (FPGA) technology allows for personalizing the system after its manufacturing to fit a specific application without loss in performance. Reconfigurable ECT systems based on FPGA technology were proposed in the literature [14–17]. These systems can be updated or reprogramed using external FPGA development tools in a JTAG mode, which requires the system to be stopped and restarted. The static JTAG reprogramming is very inconvenient and time-consuming in the case of a modular system with multiple FPGAs installed.

The data acquisition hardware developed at our laboratory is modular and has a star topology [18]. It consists of multiple boards, each equipped with FPGA chips. A central FPGA, installed together with an ARM processor on the control board, is connected to many FPGAs installed on the read-out boards. A run-time partial reconfiguration mechanism was elaborated to simplify the device configuration process. Our custom FPGA support system (FSS) allows device reconfiguration on the fly without external development tools, even to the end-user.

In this paper, the mechanism of a run-time reconfiguration of ECT data acquisition hardware is presented for the first time. The main element of this mechanism is a partial reconfiguration of FPGA performed by exchanging the code of a PicoBlaze soft-core processor. A second PicoBlaze soft-core processor acts as a reconfiguration controller and controls code reloading. The application of a tiny processor as a reconfiguration controller is an original and cost-effective method in comparison to the application of a 32-bit MicroBlaze RISC processor for this purpose [19]. The originality of the presented work also lies in the development of a reconfiguration mechanism for a modular system with a multi-level structure.

The proposed FPGA reconfiguration method is a behavioral reconfiguration [20] in which the system behavior is modified by a PicoBlaze processor code exchange. It can be seen as an alternative method to partial reconfiguration (PR). Although the effectiveness of PR has been demonstrated in many papers, the wider application is limited due to combination of design and implementation complications. The presented reconfiguration method, which is relatively simple in comparison to demanding PR, should allow wider practical application of reconfigurable systems.

The dynamic reconfiguration using a PicoBlaze processor is also proposed. In the case of time-critical applications, the short switching time guaranteed by the proposed method is crucial.

*Related Work*

The idea of FPGA application in data acquisition systems is popular because these devices have high dynamic performance and allow the performance of fast data transfer and quite complex preprocessing in real time. Reconfiguration reduces FPGA resource requirements and simplifies a chip logic compared to a static logic, which impacts system performance. The example of an FPGA-based reconfigurable system, in which new mea-

surement protocols can be run without significant hardware modifications, is presented in [21], for example. The reconfiguration of the system is completed at the microprocessor level through the exchange of subroutines of real-time application that run on an embedded microprocessor.

Reconfiguration of the system can be carried out in various ways, including partial [22], run-time [23] or dynamic reconfiguration [24,25]. For example, in [23], a configuration bitstream is downloaded to Xilinx Virtex-4 using advanced configuration environment compact flash (ACE CF) and JTAG. In [24], bitstreams are transferred through the Xilinx ICAP (Internal Configuration Access Port) interface.

Reconfiguration controllers can have different architecture. In [26], the MicroBlaze soft-core microprocessor was used as a reconfiguration controller. In [27], a lightweight RT-ICAP controller was designed to load bitstreams from local scratchpad memory. In [28], a SelectMAP configuration interface, as an alternative to the ICAP, was used for reconfiguration of Spartan-6 based system.

The challenges of run-time and dynamic reconfiguration designs were respectively described in [29–31]. The applications of dynamic reconfiguration, where switching time is a critical parameter, include, for example, computing [32], radar signal processing [33] and cognitive radio [24,25]. The optimal placement of PR blocks in the sense of switching time and FPGA resources size are discussed in [34,35], for example. The combination of the advantages of soft-core processor application for dynamic partial reconfiguration (DPR) is presented in [36].

For a wider use of reconfiguration, a high-level interface is necessary, an example of which is presented in [37,38]. The application of Phyton language and the Xilinx PYNQ library for PR implementation is shown in [22]. In [39], the concept of a straightforward, portable and extensible open-source communication and synchronization API for FPGA reconfigurable computing platforms is reported.

In the method proposed in this paper, a code of the PicoBlaze soft-core processor is exchanged for the reconfiguration of the system. A similar idea was presented in [19], where reconfiguration was used to change the behavior of a computing system. The system had a star topology with the central MicroBlaze soft-core processor and many PicoBlaze coprocessors. Xilinx's Fast Simplex Link (FSL) interface for MicroBlaze was used to rewrite code segments of coprocessors. In our solution, PicoBlaze is used as a reconfiguration controller instead of MicroBlaze. PicoBlaze is a time predictable processor (which makes it easier to synchronize multiple processors with each other) and uses only about 150 logic cells (logic cell comprises one LUT, one multiplexer, and one register), which is much less than in the case of the 32-bit MicroBlaze processor. MicroBlaze requires around 1000 logic cells in the smallest configuration (Xilinx Spartan-6, used in our system, contains 74,637 logic cells) [40]. Reconfiguration implemented using a PicoBlaze soft-core processor has the benefit of the reduced size of required resources compared to the reconfiguration implemented using a MicroBlaze processor. The application of a time predictable processor allows for easy estimation of the reconfiguration time, which is important in the case of dynamic reconfiguration. Another advantage of our solution in comparison to the design based on MicroBlaze is the very short time of the implementation phase.

The idea of reconfigurable and upgradable hardware is essential in electrical tomography due to the possibility of adapting the sensor to various conditions and applications. An overview of FPGA technology application in process tomography systems can be found in [41]. The reconfigurable hardware for imaging techniques, such as electrical resistance tomography (ERT), electrical impedance tomography (ECT), electrical impedance tomography (EIT) and ultrasonic tomography, is discussed.

An ECT system based on reconfigurable electronics was proposed in [14]. In this system, the acquisition electronics are designed using a programmable system on chip (PSoC) and a microcontroller unit (MCU), allowing both analog and digital blocks to be configured. The authors emphasize that the whole electronic platform can be easily reconfigured to be adapted to different applications and measuring strategies.

In another example of a reconfigurable system for ECT, wireless sensor networks (WSN) and the System-on-Chip (SOC) technology were used [15]. The presented design was based on a Nios II soft-core processor, defined in a hardware description language in the Cyclone Altera's FPGA. Another version of this system with a Cyclone V Intel FPGA and ARM Cortex-A9 processor is reported in [16].

A system with multiplexed single-channel front-end electronics and a capacitance to digital converter (CDC) chip is shown in [17]. The reconfigurable hardware based on a Virtex-II FPGA has built-in procedures for image reconstruction. The system enables the reconstruction using three computationally complex algorithms, including the one-step LBP and the iterative Landweber algorithm.

Examples of the reconfigurable system were also presented for EIT. The designs were based on FPGA [42,43] or National Instruments' DAQ system [44].

The above-mentioned examples of reconfigurable ECT systems allow only static reconfiguration, which requires the system to be stopped and reprogrammed in a JTAG mode. The ECT system presented in this article enables run-time reconfiguration. Furthermore, a new agile, cost-effective reconfiguration method is proposed.

## 2. Materials and Methods

### 2.1. EVT4 Data Acquisition System

Systems for ECT based on FPGA technology have been developed in our laboratory for years. The currently developed EVT4 data acquisition system has a modular architecture (Figure 1) to allow the usage of different measurement methods with just a simple change of measurement modules. The system has a star topology consisting of one main board connected to eight read-out boards combined with eight analog boards. Every analog board is equipped with four transmitter–receiver channels giving thirty-two measurement channels at maximum. High-speed 16-bit analog-to-digital converters (ADCs) running at 10 mega samples per second (MSPS) in many channels generate a significant data stream that is challenging to transmit. The data from the ADCs are transferred to the read-out board using a LVDS serial link. The whole control logic of a read-out board is placed in a Xilinx Spartan-6 programmable logic device (FPGA). Read-out boards are connected to the mainboard using Xilinx GTP multi-gigabit transceivers, with SATA as a physical standard. The mainboard has eight SATA sockets for connections with the read-out boards and contains a Spartan-6 FPGA and an ARM Cortex-A8 processor. The Ethernet link is used to connect the mainboard with a host computer. A detailed description of the architecture is presented in [18].

The software of the EVT4 capacitance tomography system consists of the embedded software of the data acquisition system and the host computer software (Figure 1).

The EVT4 client software for data acquisition and image reconstruction installed on the host computer is written in C++ language and built using Qt libraries. The EVT4 client communicates with the evt4d server working on the mainboard using Ethernet. The commands are sent using the TCP/IP protocol. To obtain a high data transmission rate, the server sends the measurement data to the client using the UDP/IP protocol. The EVT4 multi-threaded software consists of two threads for data acquisition, a thread for online image reconstruction and a graphical user interface (GUI). The main functions of EVT4 software are control of the tomograph, data acquisition, reconstruction and visualization of the images.

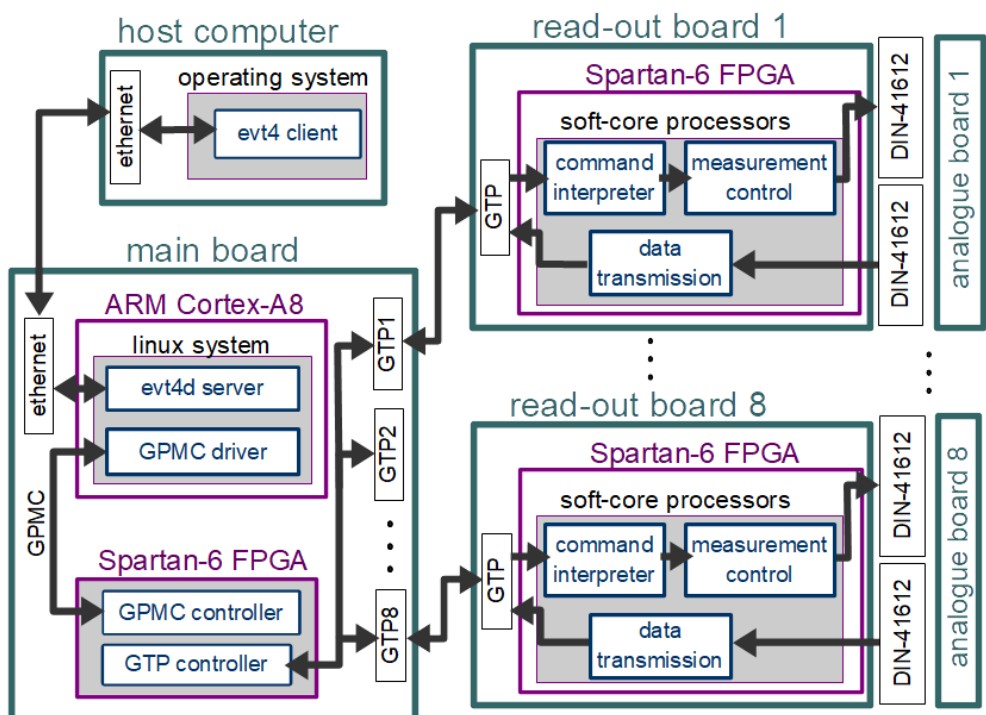

**Figure 1.** Software architecture of EVT4 capacitance tomography system.

*2.2. Embedded Software*

The embedded software consists of several parts distributed between all system modules. The software on the mainboard is divided between a C language code for the ARM Cortex-A8 microprocessor and a VHDL code for the FPGA chip Spartan-6.

Two programs are added to the Linux system running on the ARM Cortex-A8 processor: multithread server daemon evt4d and GPMC device driver. The evt4d server is responsible for communication with the host computer over Ethernet. Data transmission between the ARM processor and the FPGA over GPMC is provided by the custom GPMC driver.

The Xilinx Spartan-6 FPGA is used for the serial transmission between the mainboard and read-out boards. The VHDL code implemented on a chip consists of the GPMC and GTP multi-gigabit transmission controllers. The GPMC controller, which gives the ARM processor access to the FPGA registers, provides communication between chips. The GTP controller allows the communication with read-out boards and two FIFO buffers (the one responsible for sending and one for receiving) generated using Xilinx IP-Core generator.

The ARM processor uses the device driver for transmission to the read-out boards. At the address mapped by GPMC, 16-bit words are written to the memory. The data saved in the FPGA registers are rewritten to the FIFO buffer, from whence they are transferred using GTP serial transmission to the similar FIFO implemented on the read-out boards. Transmission in the opposite direction proceeds in the same way.

The read-out's firmware boils down to the VHDL code describing modules implemented in Xilinx Spartan-6 FPGA and assembler code describing the behavior of soft-core processors. The VHDL language is used to define the following components: a GTP controller, a block managing communication with the analog board, raw data processor, two FIFO buffers and three PicoBlaze soft-core microprocessors. The GTP controller is a part of a communication interface between the mainboard and the read-out boards. As in the mainboard, FIFO blocks are created using an IP-Core generator. The first FIFO block buffer is used for storing results of measurements coming from the analog-to-digital converter or the raw data processor block. The second block is a part of the GTP interface. The raw data processor is a block responsible for computing measurement results from ADC

readings. Three PicoBlaze soft-core processors fulfill the following functions: receiving and interpreting commands from the mainboard, controlling the measurement process held on an analog board and data transmission. Every read-out board is programmed with the same code, but each has a different number granted by the mainboard. That number is critical because it allows the read-out board to figure out to which electrodes the channels of the steered analog board are connected.

Soft-Core Processors on Read-Out Board

PicoBlaze is a very simple 8-bit soft-core processor implemented using logic synthesis in an FPGA chip. It uses only about 150 logic cells (logic element comprises one LUT, one multiplexer and one register). (Xilinx Spartan-6 used in the EVT4 device contains 74,637 logic cells.) The PicoBlaze soft-core processor was selected because of its simplicity (in terms of programming) and good performance in solving simple tasks. The PicoBlaze processor is programmed using an assembly language. Each instruction executes in two clock cycles, which makes the time needed to execute the code easy to calculate.

The schema of read-out board software architecture is shown in Figure 2. Three PicoBlaze soft-core processors clocked with 100 MHz frequency are used.

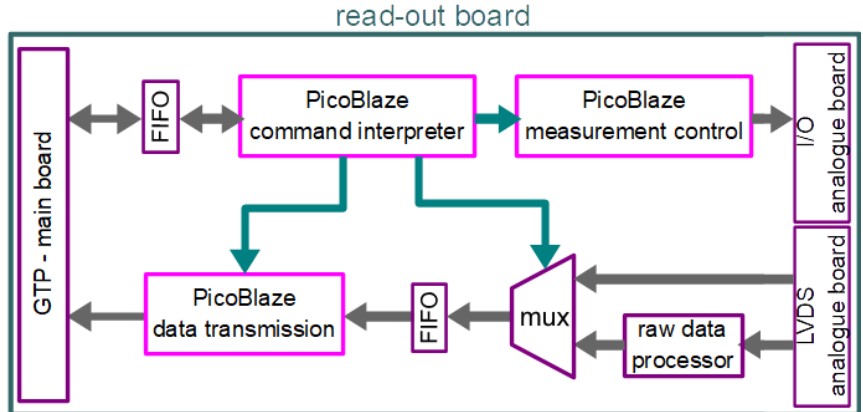

**Figure 2.** Blocks of VHDL code at read-out board.

The command interpreter PicoBlaze receives commands from the mainboard. This processor can interpret four different commands: set a number, make a measurement, set raw data flag and reset. The mainboard sends a unique number for each read-out board at boot time. The assigned number (from zero to seven) depends on the order in which cards are connected to the mainboard. The command interpreter PicoBlaze is linked with the other two soft-core processors with 8-bit signal lines. When the processor receives a measurement command, it activates the trigger bit on the line connected to the measurement control PicoBlaze. A trigger starts the measurement for one excitation with the settings specified by value on other bits of the communication line. Then, the command interpreter sends the request to the data transmission PicoBlaze to start the procedure of sending measurement data to the mainboard. Measurement results can be stored in a FIFO buffer as unprocessed ADC readings or after computing using the raw data processor block. The set raw data flag command controls the settings of the multiplexer responsible for selecting which type of data will be sent to the mainboard. Reset clears the data stored in FIFO buffers.

The second soft-core processor is responsible for managing the measurement process by controlling switches on the analog board. The switches determine which channel works as the excitation circuit and which channels act as measuring circuits of the tomographic sensor. Other switches adjust the levels of the gains of the amplifiers in the measuring circuit. The table of the gains for the entire measurements cycle is stored in PicoBlaze's scratchpad memory.

The last PicoBlaze reads the data from four channels stored in the FIFO buffer, encapsulates them into frames with a special header, and transfers them over the GTP interface to the mainboard as 16-bit signed integers.

### 2.3. PicoBlaze with a Reprogrammable Code Memory

The redesign of the default PicoBlaze structure and application of random access memory (RAM) instead of read-only memory (ROM) is proposed to allow the replacement of the program code stored in a memory segment from the outside. The second PicoBlaze processor is used as a reconfiguration controller to reload the program code of the target processor. The simplified schema of the reconfiguration block allowing PicoBlaze instruction storage reloading is shown in Figure 3. The redesigned PicoBlaze processor uses connected RAM in the same way that it uses the default type of instruction storage. Consecutive instructions are fetched from RAM by setting the corresponding value on the 'address' line. The processor is connected only to the data output port of the RAM block. It was assumed that a self-modifying code is not required, so the processor can only read from this memory block.

**Figure 3.** Reconfiguration block providing access to the PicoBlaze program code memory.

The reprogramming PicoBlaze processor is connected to the data input port of the RAM block, discussed above, and has control over the program code of the target PicoBlaze. The reprogramming processor activates the 'reset' line of the programmable PicoBlaze to stop its operation for the duration of the reconfiguration procedure. By controlling the settings of the memory 'address' line, the processor can access each stored instruction and change it. Replacement of the selected data stored in the RAM block occurs after activation of its 'write_enable' line for at least one clock cycle.

### 2.4. Reconfiguration Procedure

Although universal, the above design of the reconfiguration block was dedicated for a data acquisition system, described later. In this system, one of the PicoBlaze processors

serves as a command interpreter and reconfiguration controller. This processor receives the command with reprogramming data from the mainboard through the GTP port (Figure 1). The reprogramming data consist of two bytes with an address of the first changed instruction, two bytes with a number of transmitted instructions and a bitstream with instructions. The second (reprogrammed) processor acts as a controller of a front-end measurement circuit. This processor is equipped with RAM and its program is subject to change.

When the mainboard sends the reconfiguration command, the PicoBlaze processor responsible for interpretation starts the reprogramming procedure. The PicoBlaze code responsible for reprogramming is shown in Figure 4. The code consists of three sections marked with labels: 'start', 'programming' and 'end'.

```
start: ;---------------
       LOAD s0, FF
       OUTPUT s0, reset
       CALL read_gtp
       LOAD sB, sA
       CALL read_gtp
       LOAD sC, sA
       CALL read_gtp
       LOAD sE, sA
       CALL read_gtp
       LOAD sD, sA
programming: ;---------------
       CALL read_gtp
       OUTPUT sA, instruction_1
       CALL read_gtp
       OUTPUT sA, instruction_2
       CALL read_gtp
       OUTPUT sA, instruction_3
       OUTPUT sB, address_1
       OUTPUT sC, address_2
       LOAD s0, FF
       OUTPUT s0, write_enable
       LOAD s0, 00
       OUTPUT s0, write_enable
       ADD sB, 01
       ADDCY sC, 00
       COMPARE sB, sD
       COMPARECY sC, sE
       JUMP NZ, programming
end: ;---------------
       LOAD s0, 00
       OUTPUT s0, reset
```

**Figure 4.** PicoBlaze reprogramming code.

In the beginning, the command interpreter activates 'reset' lines of other PicoBlaze processors to stop their work. Next, it loads the start address in the registers sB and sC, which are reserved for the instruction address pointer. Then, the processor fetches the total number of the PicoBlaze code instructions, which will be transmitted during the ongoing reconfiguration procedure, and stores this number in the sE and sD registers. The 'read_gtp' procedure loads the received data byte to the sA register. The maximum size of the PicoBlaze machine code is 4096 instructions. To save time needed for transmission, there is a possibility to send only part of the code.

In the programming section, the command interpreter fetches three bytes of one instruction from the mainboard and sends it to the RAM data input port ('data_in_port' in Figure 1). The address of the current instruction is determined by the value of the address pointer registers. The PicoBlaze needs three 8-bit ports ('instruction_X') to output one instruction and another two ports ('address_X') to control the address line of RAM. The command interpreter activates the RAM 'write_enable' line for four clock cycles. As a result, the program code stored in RAM is replaced by the new instruction. Then, the processor increments the address pointer and compares it with the expected total number

of instructions. If the pointer value is lower than the address stored in sD and sE registers, the processor jumps to the beginning of the loop ('programming' label). Otherwise, the processor goes to the last section of the code ('end' label).

In the end, the processor changes the reset lines to low and jumps to the main program loop. As a result, all soft-core processors return to the normal operating mode.

### 2.5. Dynamic Reconfiguration

The dynamic reconfiguration can be achieved using an extension of the presented method with a reconfigurable PicoBlaze connected to more than one RAM (Figure 5). The controller code for dynamic reconfiguration is shown in Figure 6. The reconfiguration controller swaps the processor code using a multiplexer in this mode. The swapping procedure can be triggered using an interrupt signal. An interrupt event forces the processor to execute the call to the last program memory address. The instruction at this address is a jump location to an interrupt service routine (ISR). The programmed PicoBlaze is stopped by the 'reset' signal, active memory block is selected and PicoBlaze is started by the 'reset' signal deactivation. This moves the instruction pointer at the beginning of the program. The reprogrammed PicoBlaze will execute a code from another RAM. The swapping procedure itself requires only three instructions (six clock cycles). Triggering the procedure with an interrupt requires executing the three additional instructions.

The reconfiguration controller can reprogram each memory block individually.

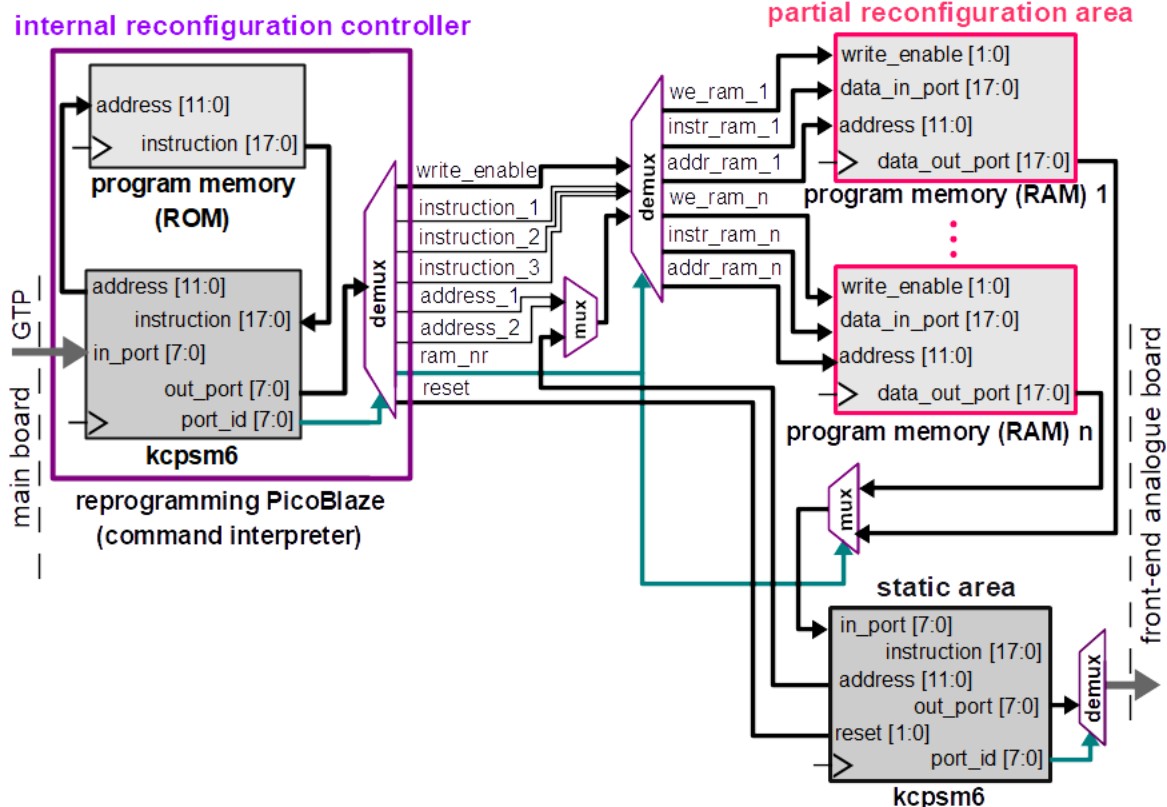

**Figure 5.** Dynamic reconfiguration block.

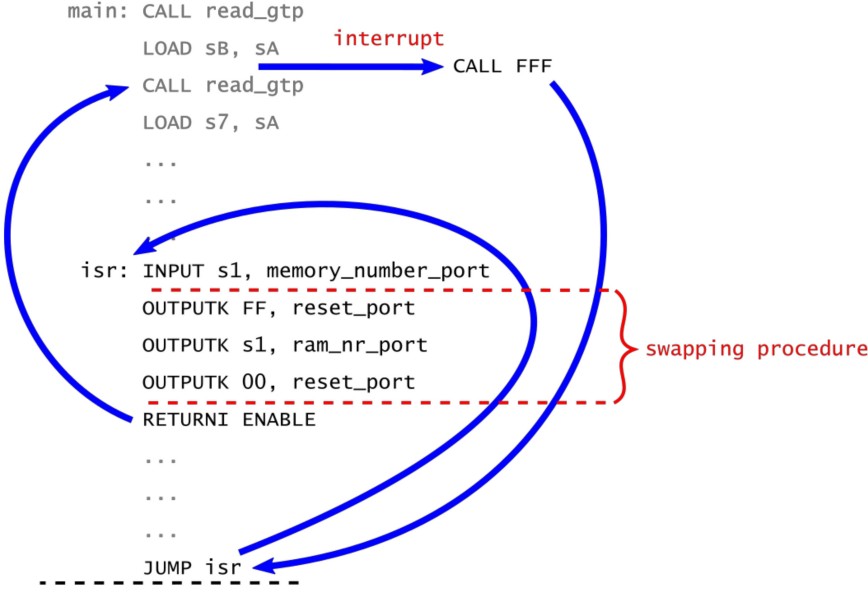

**Figure 6.** The controller code for dynamic reconfiguration. Reconfiguration triggered using an interrupt.

*2.6. FPGA Run-Time Support System Commands and Reconfiguration Procedure*

Partial reconfiguration of the EVT4 DAS consists of the reprogramming of a selected soft-core processor on the read-out board's FPGA. The elements of the reconfiguration mechanism (FPGA support system—FSS) are the following: a block providing access to the PicoBlaze program code memory on the read-out board chip, communication commands at a different level of the system structure and machine code generation procedures on the host computer (Figure 7).

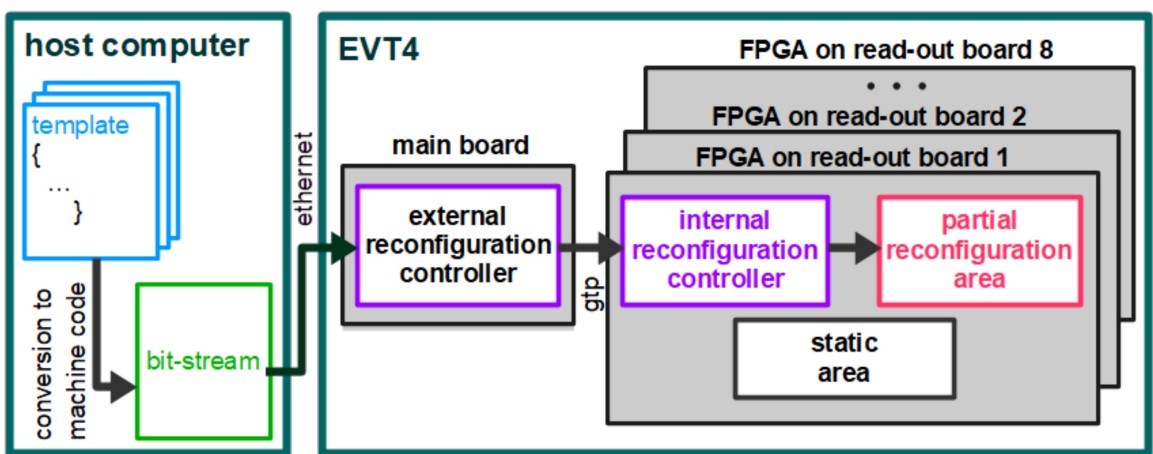

**Figure 7.** Elements of reconfiguration mechanism.

The reconfiguration command is a part of an upper layer communication protocol between the host computer and the EVT4 DAS. The reconfiguration command is sent in JavaScript Object Notation (JSON), which consists of four attribute–value pairs. Figure 8 shows a description of this command prepared in a JSON Schema format. The values of the 'command' and 'picoblaze' pairs are strings. Using the second string, the application on the mainboard recognizes which PicoBlaze processor should be reprogrammed. The 'offset' value is an address of the program block to be replaced. The last pair with the 'programCode' attribute contains a PicoBlaze machine code (sent as character data).

```
{
    "type": "object",
    "required": ["command", "picoblaze", "offset", "programCode"],
    "properties": {
        "command": {
            "type": "string",
            "description": "command name. 'amplifierControl' for gains reconstruction"
        },
        "picoblaze": {
            "type": "string",
            "description": "name of the roconfigured PicoBlaze. 'meas_ctrl' or 'send_data'"
        },
        "offset": {
            "type": "integer",
            "description": "16-bit address of the first program block to be reconfigured"
        },
        "programCode": {
            "type": "string",
            "description": "PicoBlaze machine code as a hex string"
        }
    }
}
```

**Figure 8.** Structure of upper layer communication protocol command for reconfiguration.

An example of the reprogramming command used in the communication between the mainboard and eight read-out boards is shown in Figure 9. The command header contains the information on which the soft-core processor is selected to be reprogrammed and the number of code instructions that will be updated. New machine code instructions are sent in 24-bit packages.

```
{   'command' : 5,    8-bit                // command header
    'picoblazeNumber' : 1,    8-bit  // which PB has to be reprogrammed
    'offset' : 0,    16-bit
    'numberOfInstructions' : 4096,    16-bit
    for (i = 0; i < 4096; ++i) {
        'padding' : 0,    6-bit
        'instruction' : data_bits,    18-bit}
}
```

**Figure 9.** Structure of reconfiguration command sent from the mainboard to the read-out boards. Reprogramming of the control PicoBlaze processor.

Different template files of a source code are stored at the host computer. Each template file is responsible for a different measurement sequence (and depends on the number of electrodes and the sensor layout). A modification or exchange of the code in the template is possible. The evt4 client software allows for selection between templates prepared for different sensors. Even the user unfamiliar with the architecture of the device can choose the values of the gains in the measuring circuit. The software replaces settings stored in the template files with those selected in the interface. On the other hand, the firmware developer who knows specific PicoBlaze assembly language can edit source code directly in the template file to change the whole measurement structure or add new functionalities to the device.

The selected template file is converted into machine code using a kcpsm6 assembler. Then, the host computer attaches the resulting bitstream to the reconfiguration command

and sends it to the main board of the EVT4 hardware. When the reconfiguration command is detected by the mainboard, the bitstream with a proper command header is broadcast to every connected read-out board. The command interpreter PicoBlaze on the read-out boards starts the reprogramming procedure. Finally, the bitstream with a machine code is inserted into the program memory at the specified offset.

## 3. Results

The elaborated mechanism of run-time reconfiguration was tested using the EVT4 DAS described above. Two code templates for the control PicoBlaze processor were prepared. The first template included an algorithm for a tomographic scan using a 2D tomographic sensor (Figure 10a) which had 16 electrodes in one ring. The second template contained an algorithm for a tomographic scan using a 3D sensor equipped with 32 electrodes spread over two rings (Figure 10b). In ECT, the tomographic scan consists of cycle of stimulations. A selected electrode is a stimulating electrode and others are sensing electrodes in one stage of the cycle. The number of stages in the tomographic scan equals the number of electrodes in the sensor.

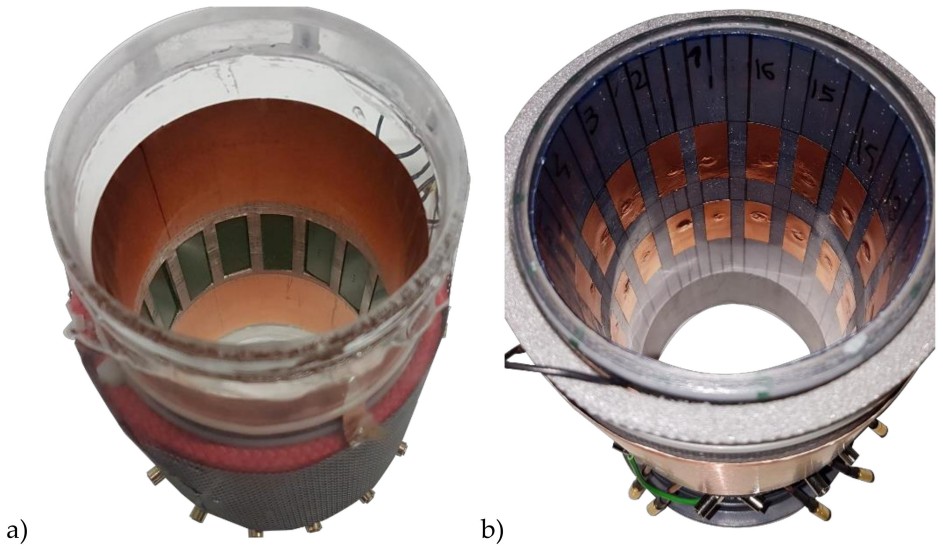

a)                                           b)

**Figure 10.** The ECT sensors used in the experiments: (**a**) 2D with 16 electrodes, (**b**) 3D with 2 rings of 16 electrodes each.

In an ECT sensor, the capacitance value of the opposite electrodes is smaller than for the adjacent electrodes by two or even three orders of magnitude. The gain adjustment is required for the measurement values to fall within the measuring range of the input circuits and to increase the signal-to-noise ratio [13]. The gains for all electrode pairs were selected for both sensors using the graphical user interface of the EVT4 client application (Figure 11). The software enables the selection of the gain in the function of the distance between the stimulating and sensing electrode. The estimated signal value is plotted on the graphs for both the sensor filled with low permittivity material and the sensor filled with high permittivity material. Due to the circular symmetry of the cylindrical sensors, the same gain table was used for each measurement cycle with the different stimulating electrode. For typical sensors, the gains are increased for pairs of opposite electrodes whereas they are decreased for pairs of adjacent electrodes. The software automatically adds the gain levels and corresponding processor instructions to the template file. These instructions are related to reading the gains from the gain table stored in the PicoBlaze's scratchpad memory. In the scan algorithm, the value of the gain in the measuring channel is adjusted at each cycle using the information about the distance of electrodes. The gain table in which the position corresponds to the distance between the excitation and sensing electrode was used. An initial rotation of this table depends on the read-out board number.

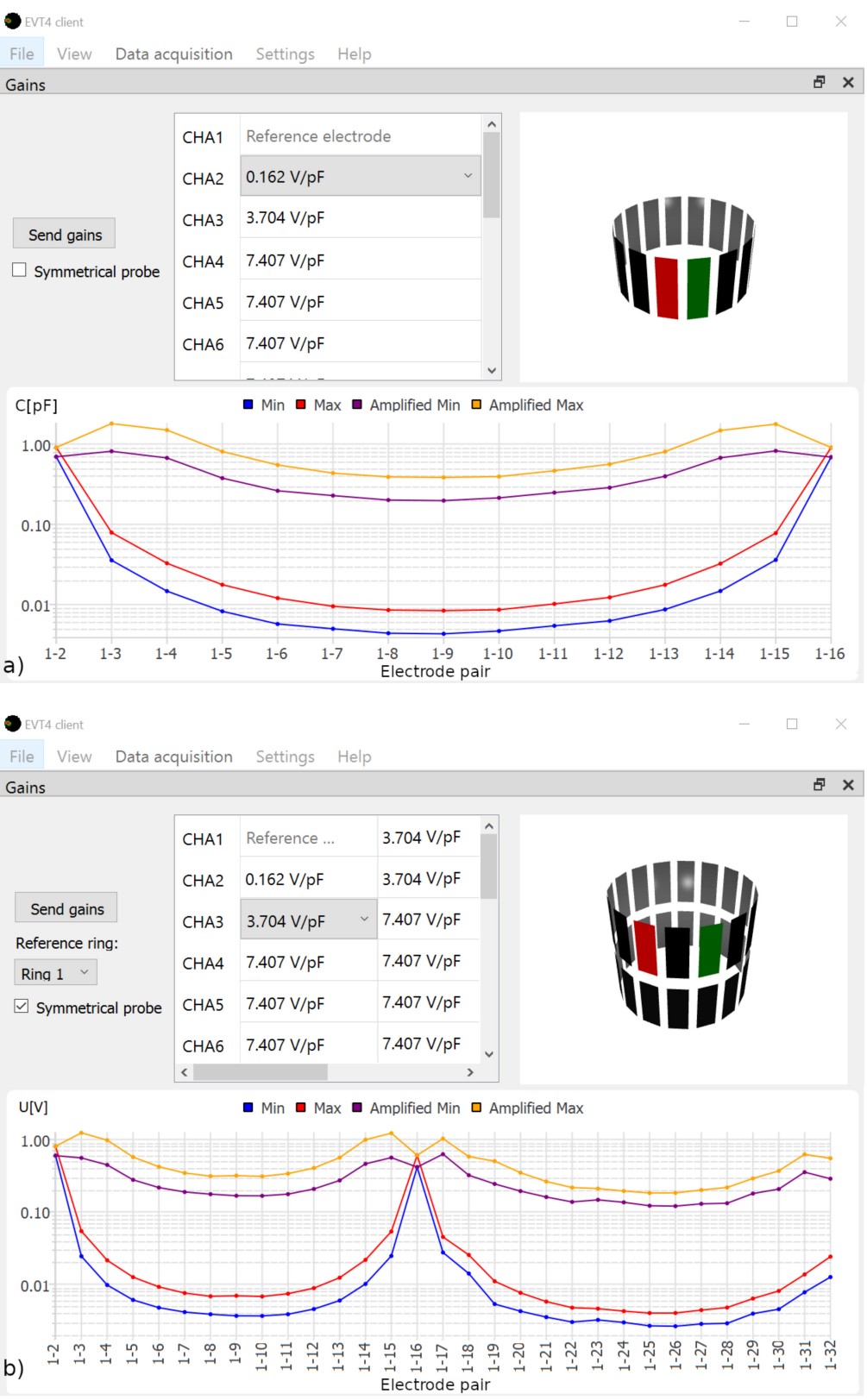

**Figure 11.** EVT4 client graphical user interface for gain adjustment for: (**a**) 2D sensor; (**b**) 3D sensor. The stimulating electrode is marked in red on the sensor model. The green electrode is the sensing electrode for which the gain is being set. The estimated signal value at low gain for the sensor filled with low (Min) and high (Max) permittivity material. The estimated value at high gain (Amplified Min, Amplified Max). The high gain is set for all electrode pairs except the adjacent electrodes.

The configurations' bitstreams for 2D and 3D sensors with two different gains were prepared for the experiments. The data acquisition system was reconfigured using the developed run-time reconfiguration mechanism. The tests confirmed correct measurements using 2D and 3D sensors. The ADC readings of capacitance values for electrode pairs increased as expected with the gain increase for both sensors (Figure 12).

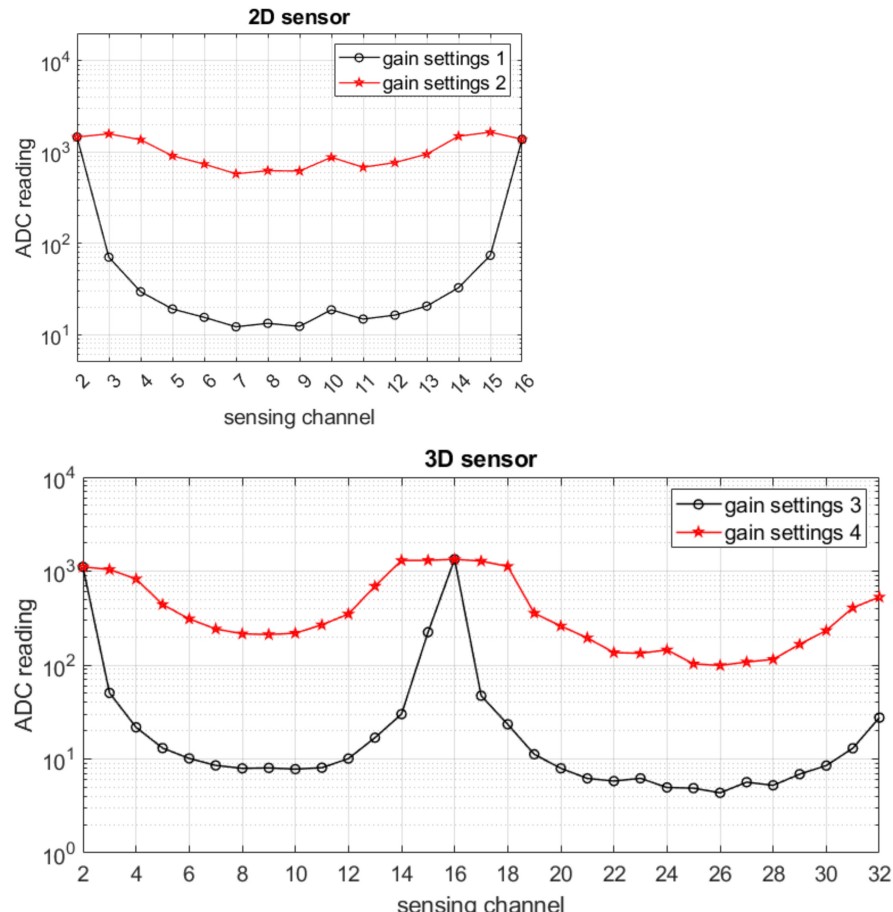

**Figure 12.** ADC readings in a sensing channel number (pair of electrodes) function at different gains. Measurement for 2D (**top**) and 3D sensor (**bottom**). The first channel is connected to an excitation electrode. All gains in settings 1 and 3 are equal to 0.162. The gains in settings 2 and 4 are equal to 7.407 except for the channels corresponding to the adjacent electrodes (0.162 for immediately adjacent electrodes; 3.704 for electrodes distant by two electrodes).

The test object was measured using the 2D and 3D sensors incorporating arbitrarily selected gain settings. The object was made of PLA using 3D printing and consisted of three cylinders (Figure 13a). Each cylinder had a different position in the XY plane and the Z axis. The diameter and height of cylinders were equal to 20 mm. The sensing domain (inside of a sensor) was a cylindrical volume with a diameter of 98 mm and a height of 25 mm for the 2D sensor and a height of 70 mm for the 3D sensor. The slice of the test object was reconstructed from 2D measurements using an iterative Landweber algorithm (Figure 13b). Two cylinders reconstructed more poorly because only a part of their volume was into the sensing domain. The slices of the 3D volume reconstructed from 3D measurements are shown in Figure 13c.

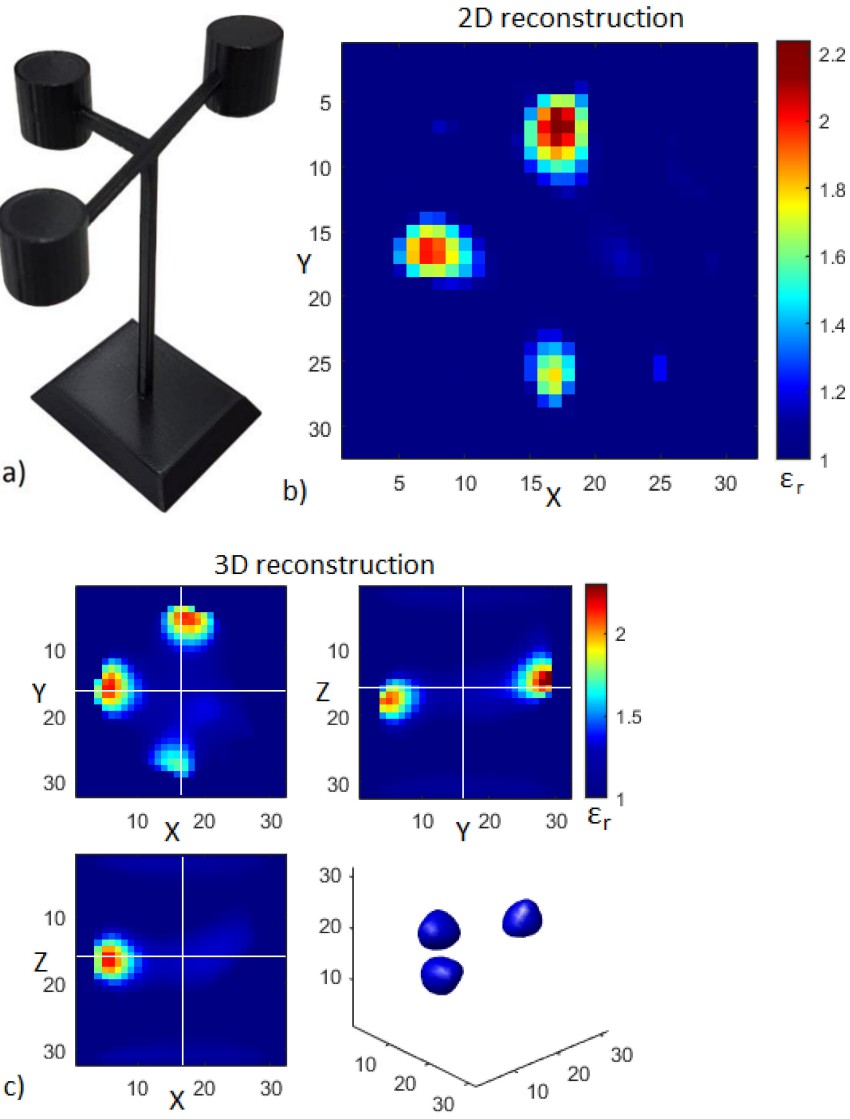

**Figure 13.** (**a**) 3D printed test object of PLA (relative permittivity 2.75). (**b**) 2D images reconstructed using the Landweber iterative algorithm. (**c**) Slices of 3D volume reconstructed using the Landweber algorithm. Crosshair markers show the mutual position of slices. The segmentation result is shown by surface shading.

The experiments allowed us to assess the parameters of the developed FPGA run-time reconfiguration procedure. The time needed to perform a reconfiguration is one of the most important parameters in dynamic applications and critical in some designs [14]. The time required to download the bitstream to the RAM of the reprogrammed PicoBlaze processor is proportional to the bitstream size. Each PicoBlaze instruction is 18 bits wide. The PicoBlaze can address up to 4096 instructions, which correspond to a maximal memory size of 73,728 bits. The time required to write this data to the RAM of the programmed processor consists of the preparation phase and bitstream loading. The reprogramming procedure's initial preparation and final completion require 30 reconfiguration controller instructions. Loading the bits of one instruction to the processor RAM requires the execution of 32 reconfiguration controller instructions. A bitrate, which describes reconfiguration speed independent of the bitstream size, can be calculated using the formula:

$$bitrate = \frac{instructions * bits\_per\_instruction}{instruction\_time * (30 + 32 * instructions)} = \frac{73.728 \text{ kb}}{2.62204 \text{ ms}} \approx 28 \text{ Mbps}$$

where *instruction_time* equals 20 ns at a 100 MHz clock used in the FPGA. The obtained bitrate, which is comparable to the bitrate reported in [30,45,46], can be sufficient for many applications; however, it may be insufficient for some dynamic applications where the required speed is higher (Table 1).

**Table 1.** Partial Reconfiguration Speed—Bitrate.

| | |
|---|---|
| PicoBlaze with reprogrammable code memory (our solution) | 28 Mbps |
| Processor Configuration Access Port (PCAP) in Zynq 7000 under Linux [45] | 78.7 Mbps full, 62.8 Mbps partial |
| 8-bit ICAP bandwidth [47] | 95.3 Mbps |
| Ethernet controller + 8-bit ICAP [30,46] | 80 Mbps |
| 32-bit ICAP bandwidth [47] | 381.5 Mbps |
| Internal controller + 32-bit ICAP [24] (estimated from other parameters) | 320 Mbps |
| Overclocked (200 MHz) ICAP port [32] | 800 MBps |
| Virtex Ultrascale [32] | 800 MBps |

The second important parameter of reconfiguration, next to the duration, is the resource size required for partial reconfiguration. The components of FSS used for reconfiguration in our system do not increase the usage of FPGA's resources significantly. Only 61 additional logic cells are required to implement the reconfigurable version of our project. The exchange of the PicoBlaze program memory type and the addition of the logic providing memory access increases the utilization of FPGA resources needed by our project by 0.6%.

The elements of the elaborated FPGA support system were tested successfully using the EVT4 system. The reconfiguration mechanism allows parallel reprogramming of eight FPGAs on the eight read-out boards. The reconfiguration is initialized by sending the reconfiguration command (Figure 9) from the mainboard to the read-out boards. The total reconfiguration time is equal to the time needed for reconfiguration on the read-out board, since the command transmission from the mainboard does not introduce delay.

In the performed test, about 300 instructions of the measurement control processor code were downloaded, a process which lasted 192.6 µs. The obtained reconfiguration time is comparable to the duration of the one stage of the tomographic scan, i.e., the duration of stimulation using one electrode. Thus, the gain modification before each stage would considerably increase the length of the tomographic scan. In the case of the initial configuration of the data acquisition system for a specific sensor, the duration of the reconfiguration procedure is negligible.

## 4. Discussion

In this paper, the run-time reconfiguration mechanism of a data acquisition system for electrical capacitance tomography was elaborated. The original mechanism allows reconfiguration, on the fly, of a modular EVT4 system with multiple FPGAs installed on separate read-out boards. The reconfiguration using this mechanism is performed without resetting the device. This solution definitely facilitates the possibility of adapting the device to any sensor and any measurement sequence, as it does not require the programming of each FPGA system individually in a JTAG mode. The communication commands developed for the reconfiguration constitute an extension of the communication layer used to control the data acquisition system.

The implementation of different measurement sequences that are necessary in ECT applications will, in one program, result in a complicated code with many parameters and many conditional statements. In general, it would be difficult or even impossible to consider all possible sensors and measurement strategies in one finite state machine (FSM) implemented using VHDL or PicoBlaze code. Instead, a simple program dedicated to a given sensor and optimized for speed can be loaded by the user at the run-time.

An EVT4 firmware developer with only general knowledge about the system architecture can change measurement structure or add new functionalities to the device on a higher level. The PicoBlaze assembly language must be known to design a new template file. The

reconfiguration block in the FPGA chip is also easy to extend with more programmable PicoBlaze processors. For example, in the EVT4 system, there is the possibility to change the program memory block of the data transmission PicoBlaze, as well. This provides an opportunity to develop more functionalities in the future.

The proposed application of the PicoBlaze soft-core processor for behavioral reconfiguration is relatively simple for a potential user, whereas the Xilinx partial reconfiguration design option has a reputation as a solution demanding expert-level skill [48]. Additionally, the introduced method of reconfiguration is not patented and does not require licensing, as does the commercial Xilinx solution.

The simple reconfiguration controller implemented using a PicoBlaze processor is an original element of the proposed method. This solution provides the effectiveness of the reconfiguration mechanism. Adding such a reconfiguration controller to the design does not noticeably increase the resource utilization of the FPGA. The obtained bitrate (28 Mbps) can be sufficient even for dynamic applications. However, it may be insufficient for time-critical applications. The added reconfigurable blocks do not significantly increase the number of necessary FPGA cells in the presented data acquisition system. In the case of a data system with many PicoBlaze processors used for data preprocessing, the increase in FPGA resources will be significant. In a computing system where many PicoBlaze processors compose a multicore accelerator, the cost of reconfiguration may be significant. The reconfigurable PicoBlaze requires about 40% more logic cells than standard PicoBlaze (150 logic cells). A multicore system with one reprogramming processor and many reprogrammed processors will have approximately 1.4 times fewer cores than a system consisting of only unconfigurable processors.

In our solution, the bitstreams used for reconfiguration do not occupy the FPGA cells. Bitstreams are stored in the memory of a host computer instead of in (local) FPGA scratchpad memory because our system does not require dynamic reconfiguration. The process of switching to the other configuration is slow, but the number of configurations is not limited by the FPGA's size.

An extension of the presented method for dynamic partial reconfiguration was suggested. In the proposed solution of dynamic reconfiguration, several bitstreams can be preloaded to the FPGA and switched dynamically. The code swapping and interrupt service subroutines require only 12 clock cycles, i.e., 120 ns at 100 MHz clock frequency. This time is independent of the code size. The given time is only part of a total response time of a real-time system that, for example, has to analyze data to react to changing conditions.

The PicoBlaze soft-core processor ensures lower performance and slower response to simultaneous inputs than native VHDL. Our PicoBlaze based method may be an option if fast reconfiguration is required but the slower FSM is satisfactory.

## 5. Conclusions

In this paper, we showed the original application of run-time reconfiguration in a data acquisition system for ECT. Our method ensures a high speed of data acquisition due to the execution of only a small part of the general algorithm at a given time interval. The originality of the presented work also lies in the application of elaborated reconfiguration in a system with a multi-level structure where the host computer controls a process of modification of code in the peripheral FPGAs.

The elaborated run-time reconfiguration method allows for switching between different configurations of the measurement channels in the ECT system. Generally, this procedure can also be used to modify the whole sensing cycle of the ECT system or to modify the data preprocessing algorithm performed by the other PicoBlaze processors. In the future, several algorithms will be predesigned in such a way that the user can easily reconfigure the system for different types of sensors and a variety of sensing procedures.

**Author Contributions:** Conceptualization, W.T.S. and D.W.; methodology, D.W., W.T.S. and J.K.; software, J.K. and D.W.; validation, J.K, P.W. and M.M.; formal analysis, W.T.S. and D.W.; investigation, D.W.; resources, P.W. and M.M.; data curation, J.K. and D.W.; writing—original draft preparation, W.T.S. and D.W.; writing—review and editing, J.K, P.W. and M.M.; visualization, D.W. and J.K.; supervision, W.T.S.; project administration, W.T.S.; funding acquisition, all authors contributed equally. All authors have read and agreed to the published version of the manuscript.

**Funding:** This research was funded by Excellence Initiative—Research University grants by the Ministry of Science and Higher Education (PL), grant number 1820/11/Z01/POB4/2021.

**Institutional Review Board Statement:** Not applicable.

**Informed Consent Statement:** Not applicable.

**Conflicts of Interest:** The authors declare no conflict of interest.

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
