# Peer review of "A Run-Time Reconfiguration Method for an FPGA-Based Electrical Capacitance Tomography System"

_electronics, doi:10.3390/electronics11040545_

Round 1

Reviewer 1 Report

Literature background information in introduction section is good enough to understand the concept of the capacitance tomography system.  Software coding and system architecture of the proposed capacitance tomography system is well described in detail. There are no English grammar issues.  However, there are small simulation results for the system. Therefore, the manuscript could be revised if authors follow the questions.

  1. Figures 1-6 fonts are too small.
  2. Figure 7 code font quality looks bad.
  3. Figure 9 quality looks bad.
  4. Data availability section is missing.
  5. In Figure 9, there are no labels for x- and y-axes.
  6. Novelty in abstract section need to be emphasized.
  7. In Table 1, authors provided several parameters. However, authors do not show the simulated results for different sensing channels.
  8. Authors need to provide more simulated results for maximum clock speed for different channels.
  9. Authors need to provide some image or visualized data because this is tomography system if possible.
  10. In Figure 9, CHA2 and CHA3 parameters are different with others. Is there any reason ?

Author Response

Thank you for agreeing to review our article and your comments.

Reviewer 2 Report

This paper studies the original application of run-time reconfiguration in a data acquisition system for ECT. The method ensures a speed of data acquisition due to the execution of only a small part of the general algorithm at a given time interval. The contribution of the presented work is an application of elaborated reconfiguration in a system with a multi-level structure where the host computer controls a process of modification of code in the peripheral FPGAs.

The motivation of the work is not very clearly stated. 2D and 3D configurations should be compared and justified the choice. The introduction section should be rewritten.

The paragraph starting from line 120, mentioning that a code of PicoBlaze soft-core processor is exchanged for the reconfiguration of the system. A similar idea was presented, where reconfiguration was used to change the behavior of a computing system. The system had a star topology with the central MicroBlaze soft-core processor and many PicoBlaze coprocessors. Xilinx’s Fast Simplex Link (FSL) interface for MicroBlaze was used to rewrite code segments of coprocessors. The current work is very closely related to this work. Therefore, the novelty and solution of the paper should be discussed.

On page 4, it is mentioned in the beginning that a system with a multiplexed single-channel front-end electronics and a capacitance to digital converter (CDC) chip is shown in reference 40. The hardware based on a Virtex-II FPGA can be reconfigured for a one-step LBP or iterative Landweber algorithm. This sentence is difficult to follow. Please give a detailed explanation.

The complexity of dynamical reconfiguration should be discussed. Ideally, the time complexity should be proved. I am interested to see a rigorous proof, which is essential for a paper to be published in reputable journals like Electronics. The simulation works need to be be underpinned rigorously.

In Figure 11, would it be better if you use box plots instead? Also, some of the figures are out of place. They are too large and do not fit in the size of the page. Please provide the replacement figures.

The discussion section should be expanded. The time and resources required for FPGA should be investigated along with the proposed method.

Author Response

(The authors gave the same response as above.)

Round 2

Reviewer 1 Report

Authors answered the questions very clearly. Authors showed the answers step by step. It is really good work. Thus, I recommend authors' work to be published.

Reviewer 2 Report

The paper now looks good. I recommend acceptance.